# Genome-Wide Association Study Identified Novel SNPs Associated with Chlorophyll Content in Maize

**DOI:** 10.3390/genes14051010

**Published:** 2023-04-29

**Authors:** Yueting Jin, Dan Li, Meiling Liu, Zhenhai Cui, Daqiu Sun, Cong Li, Ao Zhang, Huiying Cao, Yanye Ruan

**Affiliations:** 1College of Bioscience and Biotechnology, Shenyang Agricultural University, Shenyang 110866, China; jinyt1115@163.com (Y.J.);; 2Key Laboratory of Soybean Molecular Design Breeding, Northeast Institute of Geography and Agroecology, Chinese Academy of Sciences, Changchun 130102, China; 3Liaoning Province Research Center of Plant Genetic Engineering Technology, Shenyang Key Laboratory of Maize Genomic Selection Breeding, Shenyang 110866, China

**Keywords:** GWAS, chlorophyll content, area under the chlorophyll content curve, single-nucleotide polymorphisms, candidate gene prediction, maize (*Zea mays*), ear leaf

## Abstract

Chlorophyll is an essential component that captures light energy to drive photosynthesis. Chlorophyll content can affect photosynthetic activity and thus yield. Therefore, mining candidate genes of chlorophyll content will help increase maize production. Here, we performed a genome-wide association study (GWAS) on chlorophyll content and its dynamic changes in 378 maize inbred lines with extensive natural variation. Our phenotypic assessment showed that chlorophyll content and its dynamic changes were natural variations with a moderate genetic level of 0.66/0.67. A total of 19 single-nucleotide polymorphisms (SNPs) were found associated with 76 candidate genes, of which one SNP, 2376873-7-G, co-localized in chlorophyll content and area under the chlorophyll content curve (AUCCC). *Zm00001d026568* and *Zm00001d026569* were highly associated with SNP 2376873-7-G and encoded pentatricopeptide repeat-containing protein and chloroplastic palmitoyl-acyl carrier protein thioesterase, respectively. As expected, higher expression levels of these two genes are associated with higher chlorophyll contents. These results provide a certain experimental basis for discovering the candidate genes of chlorophyll content and finally provide new insights for cultivating high-yield and excellent maize suitable for planting environment.

## 1. Introduction

Maize (*Zea mays* L.) is an important source used for animal feed and bioenergy and has become the top crop produced in China, with production estimated up to 272.6 million tons (MT) in 2021 (http://www.stats.gov.cn, accessed on 20 April 2022). High yield based on photosynthesis has always been the focus topic in crop breeding [1]. Chlorophyll is essential for photosynthesis and promotes light energy absorption and assembly with photosynthetic protein complexes [2]. Chlorophyll content is one of the most important physiological traits, which is closely related to leaf photosynthesis and crop yield potential [3]. In a certain range, chlorophyll content is positively correlated with photosynthetic rate, which directly determines the yield of crops [4]. Therefore, maintaining high chlorophyll content in leaves can improve photosynthetic activity and increase yield [5,6]. At present, chlorophyll content has been used to evaluate the photosynthetic capacity and yield potential of leaves in rice [7]. Therefore, a better mining of the genes of chlorophyll content in maize leaves is of great value to speed up maize high-yield breeding.

Chlorophyll content is controlled by nuclear genes [8], and chlorophyll metabolism can be divided into four major steps [9]. The first step is the synthesis of chlorophyll-a via the branched tetrapyrrole biosynthesis pathway [10,11]. The second step is catalyzing the mutual conversion of chlorophyll-a and chlorophyll-b [12]. The third step involves the degradation of chlorophyll-a via the pheophorbide a oxygenase (PAO)/phyllobilin pathway [13]. The final step is the chlorophyll cycle pathway [14]. Although significant progress has been made in research on chlorophyll metabolism, the molecular mechanism of chlorophyll metabolism is quite complex, and chlorophyll content is a quantitative trait [15] that can be used for quantitative trait locus analysis and identification. Therefore, it is crucial to deeply study the expression and regulation of genes involved in the regulation of chlorophyll metabolism.

In recent years, researchers have identified some quantitative trait loci (QTLs) of chlorophyll content in leaves of different populations of various crops from different angles and have made considerable progress [7,16,17,18,19], which has laid the foundation for further clarifying the molecular genetic mechanism determining chlorophyll content. However, the method of QTL mapping can only analyze the gene effects of differences between the parent materials of the isolated population and cannot widely excavate the genes regulating chlorophyll content in the whole genome [20,21]. Genome-wide association study (GWAS) is an efficient method developed in recent years to study complex traits. It has the advantages of high resolution and throughput, which can associate multiple complex traits and detect multiple alleles at the same time. It was previously reported that GWAS was used to mine candidate genes related to chlorophyll content and tolerance of soybean cyst nematode, and 15 candidate genes related to tolerance of soybean cyst nematode and chlorophyll content were identified [22]. Dhanapal et al. [23] conducted GWAS of soybean chlorophyll traits based on canopy spectral reflectance and leaf extracts and found 15 SNPs loci related to total chlorophyll content. Herritt et al. [24] identified 21 chlorophyll fluorescence phenotypes by GWAS and found relevant genes involved in photosynthesis and electron transport. Therefore, it is important to identify genes associated with chlorophyll content. The chlorophyll content of the first maize leaf was analyzed by GWAS at the seedling stage, and two genes potentially controlling chlorophyll content in maize were identified, a homolog of the *Arabidopsis Tic22* and a homolog of rice *SAG12* relating to aging [25]. However, no studies have been conducted on chlorophyll content and AUCCC of maize ear leaves within multi-year, multi-location trials.

As a research tool, GWAS have become a common way to study natural variation and inheritance of important agronomic traits in various plants [26]. Maize has rich genetic diversity and rapid linkage disequilibrium decay, which make maize an excellent variety for GWAS. To date, many researchers have contributed to our understanding of maize through GWAS. Wang et al. [27] carried out a study on drought tolerance in maize seedlings that found *ZmVPP1*, and transgenic maize with enhanced *ZmVPP1* expression exhibited improved drought tolerance. Sun et al. [28] identified candidate genes that affect bracing root angle and diameter. Li et al. [29] used GWAS to reveal the candidate gene of maize seed germination traits and found 58 genetic variation sites and 36 candidate genes, which provided important implications for the molecular breeding of maize seed germination. These studies indicate that GWAS is a reliable tool to study the chlorophyll content of maize and to find candidate genes.

In this study, 378 maize inbred lines and 96,726 SNPs were used for the GWAS of the chlorophyll content in two environments. The purpose of this study is to identify candidate genes of chlorophyll content. Our findings provide new insights for mining candidate genes of leaf chlorophyll content.

## 2. Materials and Methods

### 2.1. Plant Materials

An association panel comprising 378 diverse maize inbred lines from the northeast of China, temperate region in the United States and the International Maize and Wheat Improvement Center (CIMMYT), Mexico, were used for GWAS. All materials were kept in the College of Bioscience and Biotechnology, Shenyang Agricultural University.

### 2.2. Field Experiments

The 378 inbred lines were grown in Fushun City, Liaoning Province, China (121°74′ E, 42°14′ N) in May 2017 (17FS) and Ledong city, Hainan Province, China (108°39′ E, 18°24′ N) in November 2017 (17LD). The field experiment was designed as random blocks with 2 replicates. Fifteen plants were planted in a 2.5 m long row with 0.6 m row distance, and the planting density was approximately 45,000 plants/ha. The inbred lines were labeled before pollination for standard field management. Chlorophyll content was measured using a portable chlorophyll meter (SPAD-502, plus Konica Minolta, Tokyo, Japan), which was non-destructive, fast and cheap. SPAD-502 readings were taken from five plants per plot on five dates at a 5 d interval starting 0 d after silking. There were three measurements in the middle of the ear leaf for each plant, and the average value was used for the statistical analysis. AUCCC was calculated based on SPAD-502 readings on all measure dates. Larger AUCCC values represent higher chlorophyll content, and lower AUCCC values represent lower chlorophyll content. The formula used to calculate AUCCC was modified from the AUCCC formula [30]:$$\mathrm{AUCCC}\;=\;[(\gamma_{i}+\gamma_{i+1})/2]\;(t_{i+1}-t_{i})$$
in which *n* is the number of assessment times, *γ* is the meter reading, *i* is the *i*th rating date, and *t* is time (in days).

### 2.3. Statistical Analysis of Phenotypes

The “PROC MIXED” program in SAS software was used to analyze the variance heritability of the phenotypic values at the two places. The mixed linear model (MLM) was used for the analysis, and the model *yijk = μ+e_l_ + rk(l) + fi + (fe)il + εlik*, where *yijk* was the phenotypic value of the attenuation rate *ijk* in this test, *μ* denoted the average value of attenuation rate, *el* was the influence of two environments, *rk(l)* was the repeated effect in the environment, *fl* was the genetic effect of the *i*th family, *(fe)il* was the interaction between genetic and environmental effects, and *εlik* was a random error.

The generalized heritability formula is: *h^2^ = σg^2^/(σg^2^ + σge^2^/e + σε^2^/re)* [31], where *h^2^* represents the generalized heritability of the trait, *σg^2^* is the genetic variance of the trait, *σge^2^* represents the variance of the interaction between genetic and environmental effects, *σε^2^* denotes the residual error, and *e* and *r* denote the number of environments and the number of repetitions of the trait in each environment. The total phenotype of the two trials was predicted and expressed by the best linear unbiased prediction (BLUP) value so as to minimize the environmental impact.

### 2.4. Genome-Wide Association Mapping

The 96,726 SNPs (MAF ≥ 0.05) were used to conduct GWAS by combining the data from two genotyping platforms (RNA-seq and SNP array). The mixed linear model (MLM) calculation method was used to analyze the association of chlorophyll content in R 4.0.3, where population structure and kinship were fitted to control false positives [32,33]. We used the standard Bonferroni correction threshold α= 1 as the significance node. The *p* value was calculated as 1/*n* (*n* = 96,726), and we obtained *p* < 1.03 × 10^−5^ as significant nodes.

### 2.5. Prediction of Candidate Genes

The most significant SNP was selected to represent the locus associated with chlorophyll content in the same LD block (r^2^ < 0.2). The physical locations of significant SNPs were determined using the B73 RefGen v4 database. The annotated genes were searched within the 100 kb region around (50 kb upstream and 50 kb downstream) the detected significant SNP and were identified based on functional domains. The function of annotation genes referred to the MaizeGDB (https://maizegdb.org/), NCBI (https://www.ncbi.nlm.nih.gov/gene) and the homologous genes of Arabidopsis.

### 2.6. Heat Map of Candidate Genes Expression

The expression amount of candidate genes in leaves of different days after maize silking was obtained from the Sequence Read Archive database of NCBI, and the mapping method was as described in [34].

### 2.7. RNA Extraction and RT-qPCR

The SNP allele effect was analyzed using R 4.0.3, and leaves of 6 varieties (Liao7980, A801, 29MIBZ2, PHVA9, LX9311 and Dan330) were selected during maize V4 stage. A leaf segment of 2 cm in length was excised from the middle of the fourth leaf and stored at −80 °C. The RNAprep Pure Plant Kit was used to extract the RNA (TIANGEN). cDNA was synthesized from 2 μg of total RNA using the FastKing RT Kit (TIANGEN). RT-qPCR reactions were performed using a Bio-Rad (Hercules, CA, USA) real-time PCR system using the SuperReal PreMix Plue (SYBR Green) (TIANGEN). Transcript levels were analyzed using the comparative CT (2^−△CT^) method [29]. *ZmTubulin1* (*Zm00001d033850*) was used as an internal control for data normalization. All data were measured in three independent biological replicates. The primers are listed in Appendix A.

## 3. Results

### 3.1. Chlorophyll Content Diversity and Heritability at Silking

Chlorophyll content was investigated at silking stage in 17FS and 17LD, and the BLUP values were calculated according to the phenotypic values of two places. The chlorophyll content of 17FS, 17LD and BLUP showed a normal distribution (Figure 1) and a wide range of values, ranging from 36.23 to 68.85 (mean 54.81 ± 4.57), 27.23 to 62.66 (mean 49.63 ± 7.89) and 44.78 to 58.61 (mean 51.96 ± 3.68), respectively (Appendix A), indicating that the chlorophyll content conforms to the quantitative trait.

The heritability of chlorophyll content is 0.67, which is medium (Table 1). There were significant and positive correlations between 17FS:17LD, 17FS:BLUP, 17LD:BLUP (*p* < 0.01; Figure 1). The Genotype × Environment interaction effects were not significant for the chlorophyll content, suggesting that the Genotype × Environment interaction effects were small and had good stability and adaptability of varieties (Table 1). These results suggested that the population’s phenotypic variations of chlorophyll content are largely affected by genetic factors, and therefore, the association panel can be used for further association mapping.

### 3.2. AUCCC and Heritability after Silking

The chlorophyll content was measured at 0, 5, 10, 15, and 20 days after silking in 17FS, and at 0, 5, and 10 days after silking in 17LD (Figure 2). We found that the chlorophyll content was the highest at silking (day 0) and gradually decreased and showed the lowest at 20 days after silking during measurement in 17FS and 10 days after silking in 17LD (Figure 2). In general, the chlorophyll content showed a downward trend. After silking, the plant enters the reproductive phase of growth. Nitrogen, phosphorus, potassium, and other nutrients are rapidly transferred to the kernel [35], and this nutrient redistribution phenomenon is most obvious from the filling stage to the mature stage of maize, which leads to the aging of maize and the decrease in chlorophyll content [36].

The AUCCC showed a normal distribution (Figure 3), and the heritability of AUCCC was 0.66, which is medium (Table 1). These results show that the AUCCC is controlled by genetics and is suitable for further GWAS.

### 3.3. Correlations of Chlorophyll Content with Other Plant Developmental Processes

The correlation coefficients (Pearson’s) were calculated between chlorophyll content of ear leaf and 10 agronomic traits, including hundred kernel weight (HKW), flowering time (FT), ear rows number (ERN), kernel number per row (KNR), ear length (EL), ear perimeter (EP), kernel length (KL), kernel width (KW), kernel thickness (KT) and kernel area (KA). As shown in Figure 4, chlorophyll content is positively correlated with HKW at the 0.05 level. Previous studies also showed that chlorophyll content was positively correlated with kernel weight in rye and barley, whereas the correlation was significant at the level of 0.01 [37,38].

### 3.4. Genome-Wide Association Analysis

To ascertain the candidate genes of chlorophyll content, we conducted a GWAS in two environments of 378 lines using each environment and BLUP values at silking as the phenotype (Figure 5). The GWAS was carried out using the MLM method with a threshold of *p* < 1.03 × 10^−5^ (Figure 5A–C and Appendix A), and a total of 15 SNPs were identified. Among them, three significant SNPs were located on chromosome 1, 7 and 10, respectively, in 17FS (Figure 5A and Table 2). Ten significant SNPs were found in 17LD, of which three were located on chromosome 1, one was mapped on chromosome 5, and chromosome 2, 4, 10 harbored 2 SNPs respectively (Figure 5B and Table 2). In BLUP, two significant SNPs were located on chromosome 2 and 5 (Figure 5C and Table 2). These results indicate that chlorophyll content in maize was controlled by multiple genetic loci in this cross-combination.

Manhattan plots of GWAS were conducted on AUCCC after silking (Figure 6 and Appendix A). There were four SNPs identified, among which two were for 17FS (Figure 6A) and two were for 17LD (Figure 6B), and there were no significant SNPs in the BLUP (Figure 6C). Notably, we identified the same SNP 2376873-7-G on chromosome 10 by GWAS for chlorophyll content at silking and AUCCC.

### 3.5. Candidate Genes

A total of 76 candidate genes were identified from 19 SNPs within the 50 kb flanking regions (Table 2 and Appendix A), of which 41 genes had functional annotations. The annotations for the candidate genes consisted of transmembrane protein, kinase, phosphatase, signal transduction protein, and transcription factors that may be involved in photosynthesis, redox, chloroplast development, and plant growth. The co-located SNP, 2376873-7-G, harbored *Zm00001d026563*, *Zm00001d026569* and *Zm00001d026574* encoding APETALA2/ethylene-responsive element binding protein (AP2/EREBP), chloroplastic palmitoyl-acyl carrier protein thioesterase and UDP-D-galacturonate, respectively (Table 2). We discovered that the genomic region containing SNP, Marker.247949, includes the gene *Zm00001d007012*, which encodes chloroplast RNA binding protein and directly participates in chloroplast morphogenesis (Table 2). In addition, ATP synthase encoded by *Zm00001d007011* can directly affect photosynthesis. Together, these results further demonstrate that the candidate genes were reliable.

### 3.6. Expression Pattern of Candidate Gene

To determine the candidate genes’ expression pattern, an in silico profiling was compiled using the published RNA-Seq datasets (Figure 7). The results showed that the expression patterns of different candidate genes varied in seven different tissues (Figure 7A and Appendix A). The color scale bar at the top of the heat map represents log10-transformed FPKM value, which represents low and high expression. *Zm00001d023314*, *Zm00001d034534*, *Zm00001d003404, Zm00001d012982* and *Zm00001d014126* had a relatively high level of expression with tissue specificity in the leaf relative to other tissues. Contrarily, *Zm00001d039221* had a relatively low level of expression in leaf (Figure 7A). *Zm00001d007011* and *Zm00001d034528* showed high expression tendencies in all tissues compared to the other candidate genes (Figure 7A). Other genes showed moderate or low expression levels in leaf, such as *Zm00001d026563* and *Zm00001d007479*, which encoded AP2/EREBP transcription factors and BSD-transcription factor, respectively (Figure 7A, Table 2 and Appendix A).

To confirm the expression pattern of the candidate genes in leaves, we performed a heat map of the candidate gene expression at 0,6,12,18,24,30 d after pollination (Figure 7B and Appendix A). *Zm00001d021162* had a low expression level at day 0, moderate on days 6 and 12, and high on days 18 and 24. *Zm00001d026563* showed high expression only on day 6. *Zm00001d003403* was lowly expressed on days 0 and 6 and moderate on days 12–30. *Zm00001d003405* was moderate on day, and low on days 6–30. *Zm00001d027601* was lowly expressed on day 0, moderate on day 6, and high on days 12–30. For *Zm00001d039221*, the expression increased gradually after pollination during measurement.

Expression of *Zm00001d003403*, encoding AMINO ACID PERMEASE (AAP), gradually increased after pollinating. AAP belongs to the amino acid/auxin permease family, which is involved in transportation of the principal nitrogen assimilates amino acid [39,40]. When the *AAP2* gene was mutated in *Arabidopsis thaliana*, the mutant leaf had more chlorophyll content than the wild-type plant [41]. The leaf chlorophyll decreased (Figure 2) with an increase in *Zm00001d003403* expression in maize. These results are in line with previous reports.

The allele variations of SNP 2376873-7-G were identified, and A allele genotypes showed higher chlorophyll content at the population level (Figure 8A and Appendix A). *Zm00001d026568* and *Zm00001d026569* genes were associated with the same SNP 2376873-7-G. The relative expressions of *Zm00001d026568* and *Zm00001d026569* were higher in the genotypes carrying the SNP A allele (Figure 8B,C). In contrast, the relative expressions of *Zm00001d026568* and *Zm00001d026569* were lower in lines carrying the G genotype (Figure 8B,C). As expected, higher chlorophyll content in the A genotypes was associated with higher expression of these two candidate genes in the selected lines shown in this study.

## 4. Discussion

Improvements in maize yield are very important to ensure world food security. Enhancing crop yield potential by enhancing photosynthesis is a major focus of modern crop breeding [42,43]. Chlorophyll content is an indicator of photosynthesis activity, which can directly affect photosynthetic efficiency [44,45,46]. A large number of functional genes related to chlorophyll have been identified by analysis of many mutants in maize, rice, and other species [47,48,49]. However, chlorophyll content is a quantitative trait controlled by multiple genes, which requires further study on its function. In particular, these genes have more functional variation, and many genomic regions remain largely elusive [50,51]. Therefore, analyzing the candidate genes of chlorophyll content and identifying its important variation sites will contribute to the improvement of maize breeding.

Compared with the traditional linkage analysis, GWAS can directly use the existing natural population as the material that has the advantage of saving time and effort [52]. However, it also has shortcomings in specific research, such as population structure that is prone to lead to false positives, linkage disequilibrium level that is affected by a variety of genetic or non-genetic factors [53,54], and the accuracy of candidate gene determination that is not enough. In order to obtain more accurate research results, we can reduce false positives as much as possible by increasing the population size. We use a mixed linear model (MLM) to take the individual kinship (K) and group structure (Q) as covariates (Q + K) at the same time, which can also effectively reduce the false positive rate [55]. Currently, GWAS approaches based on MLM are widely employed in both plant and animal systems [56]. GWAS is more and more widely used in crop research, for example, the genetic basis of several kernel-related traits and charcoal rot resistance has been reported in maize, and some candidate genes have been predicted by the GWAS [57,58].

The 378 maize inbred lines were used to construct an association population, and the candidate genes of chlorophyll content at silking and AUCCC after silking were mined by using high-density markers and phenotypic data. In this study, the heritability of the two methods was medium, and both of them conformed to normal distribution, indicating that chlorophyll content was contributed to by various types of genetic variation effects. Thee area under the curve to study dynamic processes has been widely used in the medical field; for example, the area under the curve was used to represent the dynamic distribution of the age composition of patients [59]. The AUCCC is a simple tool for breeders to evaluate the chlorophyll content. Although the heritability of AUCCC was similar to chlorophyll content in this study, Yang et al. [30] found that the heritability of the area under the curve could be effectively improved, indicating that the results of this method were reliable. As an indicator, AUCCC not only provides sufficient chlorophyll content information but can also accurately compare genotypes [60]. Furthermore, this simple manipulation method can be used to integrate applied and basic maize research, providing a good foundation for follow-up gene function studies [61].

The chlorophyll content of ear leaves was measured in two different sites, which have significant environmental differences between them; therefore, it could result in that the same SNP was not identified in 17FS and 17LD. Teng et al. [25] revealed the genetic basis of chlorophyll content of the first leaves at the seedling stage by GWAS, which identified nine SNPs. However, no same SNP was found in our study. In addition, positive correlation was detected for chlorophyll content and HKW. This is because the abundance and stability of chlorophyll in leaves significantly affect grain filling, and increasing the chlorophyll content of crop leaves can improve biomass yield and grain yield [62,63,64,65]. In general, a breeding strategy is mainly dependent on phenotypic selection of the best genotype by environmental interactions and the heritability level [66,67]. A total of 19 significant SNPs were associated with chlorophyll content at silking and the AUCCC after silking. To realize good prospects in the improvement of maize, a number of SNPs needs to be selected in order to reduce the overall time and cost.

A total of 76 candidate genes were identified for the chlorophyll content and the dynamic changes of chlorophyll content according to the position of significantly associated SNPs. In addition, Li et al. [6] previously identified gene overlap with these candidate genes, such as cytochrome c reductase. They compared transcriptome related to the physiological changes of yellow-green leaf mutant of maize and found that cytochrome c reductase participates in the tricarboxylic acid cycle. The chlorophyll content of yellow green leaves is different, which results in the difference of cytochrome c reductase. The role of cytochrome c reductase is to catalyze the transfer of electrons from coenzyme Q to cytochrome c [68], which is an important electron transporter in biological oxidation and is related to programmed cell death [69,70]. Programmed cell death leads to leaf senescence, which in turn leads to a decrease in chlorophyll content [71,72]. In addition, the co-located SNPs include *Zm00001d026563, Zm00001d026569, Zm00001d026568*, etc., which encode AP2/EREBP transcription factor 40, chloroplastic palmitoyl-acyl carrier protein thioesterase and pentatricopeptide repeat-containing protein, respectively. The AP2/EREBP superfamily is one of the largest and specific transcription factor (TF) families in plants that is involved in biotic/abiotic stress, compound storage and plant growth and development [73,74,75,76]. Chlorophyll content is an important physiological index used to measure plant growth and development [77,78]. Palmitoyl acyl carrier protein thioesterase is a key gene for de novo fatty acid synthesis [79]. Fatty acids are one of the main components of the cell membrane, and insufficient fatty acid synthesis triggers programmed cell death [80,81]. Leaf senescence is an organ-level programmed death process during plant growth and development, resulting in a decrease in chlorophyll content. In the inferior allele G, the expression of palmitoyl acyl carrier protein thioesterase is low, and fatty acid metabolism is accelerated, thus leading to a reduction in chlorophyll content. Pentatricopeptide repeat-containing protein is one of the largest protein families in plants that affects chloroplast development [82,83]. Therefore, in the dominant allele A, the chlorophyll content is high, and the expression of pentatricopeptide repeat-containing protein is high.

To explore the potential roles of candidate genes in regulating chlorophyll content, we performed expression profiling analysis on RNA-seq data from different tissues at 0, 6, 12, 18, 24, and 30 days after pollination. The expressions of AP2/EREBP transcription factor 40 and chloroplast palmitoyl carrier protein thioesterase in leaves were low, and the expressions were moderately high on the 6th day after pollination. These results indicate that they have direct or indirect effects on chlorophyll content. Thus, selection of *Zm00001d026563, Zm00001d026569, Zm00001d026568* or this SNP may help to regulate chlorophyll content in maize breeding. Taken together, the discovery of candidate genes provides help for further analyzing the molecular regulatory network of chlorophyll content on maize ear leaves. The identification of SNP will promote marker-assisted selection in maize molecular breeding.

## 5. Conclusions

In this study, we discovered candidate genes of natural variation in maize ear chlorophyll content by GWAS. Chlorophyll content appeared to be of moderate heritability and showed extensive variation in the association panel, indicating that chlorophyll content is a quantitative trait suitable for GWAS. GWAS showed that the natural changes regulating chlorophyll content had manymicro effect loci. We found 19 SNPs containing 76 candidate genes that may participate in leaf senescence, photosynthesis, and plant developmental processes. One of the SNPs was co-localized in AUCCC and chlorophyll content that was associated with five genes. *Zm00001d026568* and *Zm00001d026569* contained significant SNP and encoded pentatricopeptide repeat-containing protein and chloroplastic palmitoyl-acyl carrier protein thioesterase, respectively. They were found to be highly expressed in lines with the A allele as determined by RT-qPCR. These candidate genes provide valuable resources for further study of the molecular network regulating chlorophyll content in maize. In addition, the prediction of SNPs with chlorophyll content may help devise effective breeding plans and selection strategies to improve maize yield.

## Figures and Tables

**Figure 1 genes-14-01010-f001:**
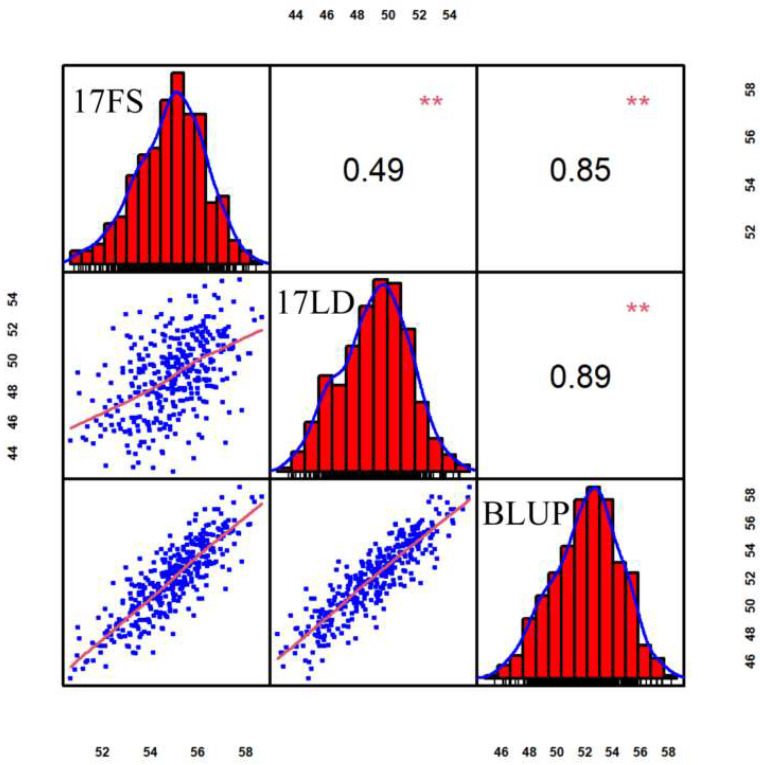
Scatterplots, frequency distribution histogram and correlation of 17FS, 17LD and BLUP. The histograms on the diagonal represent the phenotypic distribution frequency, the values above the diagonal represent the Pearson’s correlation coefficient between adjacent environments, and the scatterplots below the diagonal represent the degree of data fit. The values in the outer circle represent the range of phenotype values in the corresponding environment. ** indicates statistical significance with *p* < 0.01 significant.

**Figure 2 genes-14-01010-f002:**
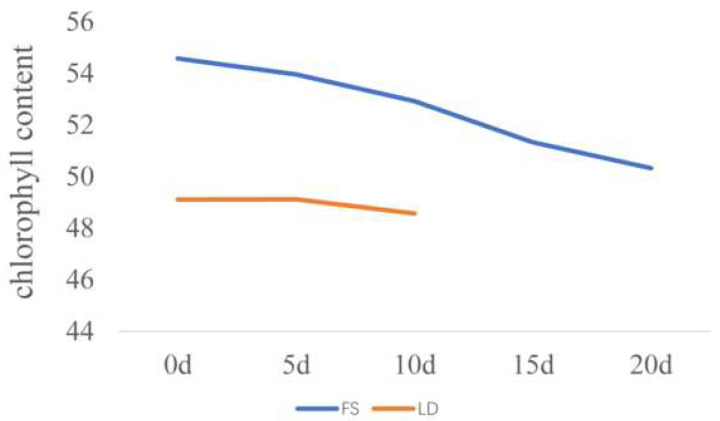
The changes in the chlorophyll content in 17FS and 17LD. The chlorophyll content was measured at 0, 5, 10, 15, 20 days in 17FS, and at 0, 5, 10 days in 17LD after silking (data are shown as the average (n = 5)).

**Figure 3 genes-14-01010-f003:**
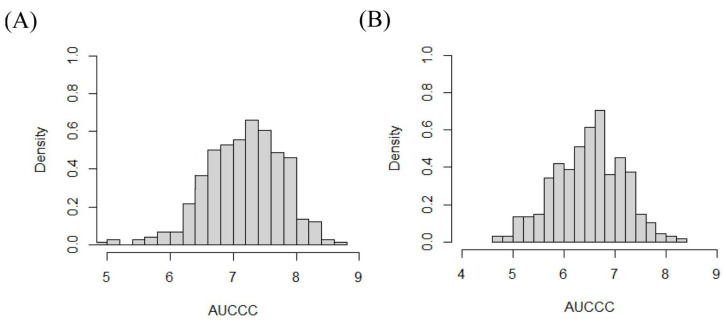
Frequency histogram of AUCCC distribution. (**A**) 17FS; (**B**) 17LD.

**Figure 4 genes-14-01010-f004:**
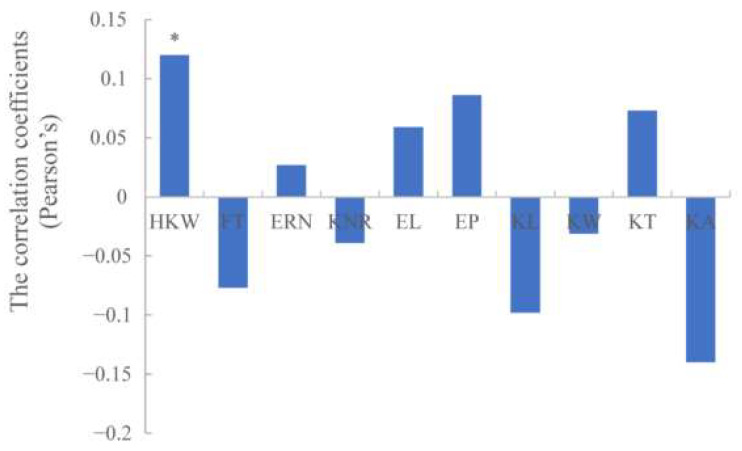
Correlation coefficients of maize chlorophyll content with 10 agronomic traits based on BLUP values. * indicates statistical significance with *p* ≤ 0.05.

**Figure 5 genes-14-01010-f005:**
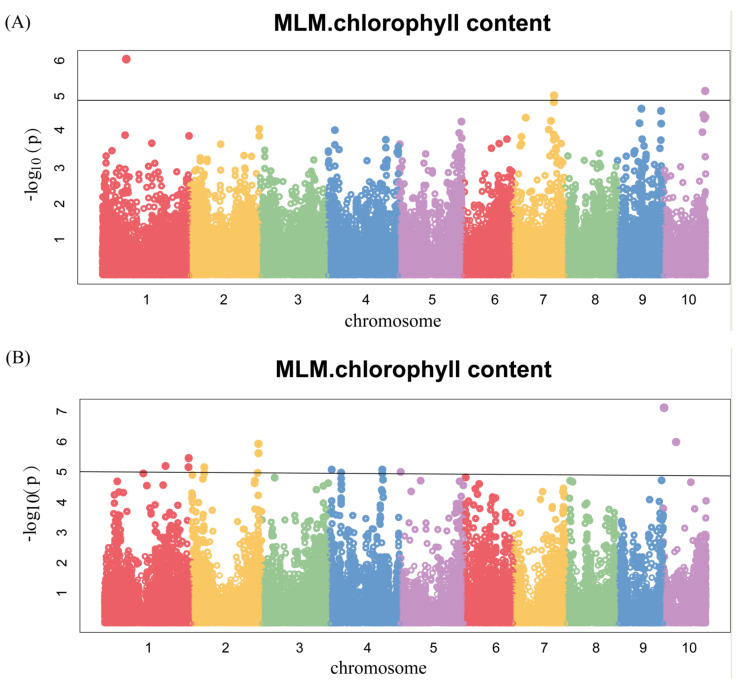
Genome-wide association studies of chlorophyll content. Each point represents an SNP site, and the horizontal black line represents that the effective threshold of Bonferroni correction at 1.03 × 10^−5^, different colors represent different chromosomes of maize. (**A**) 17FS; (**B**) 17LD; (**C**) BLUP.

**Figure 6 genes-14-01010-f006:**
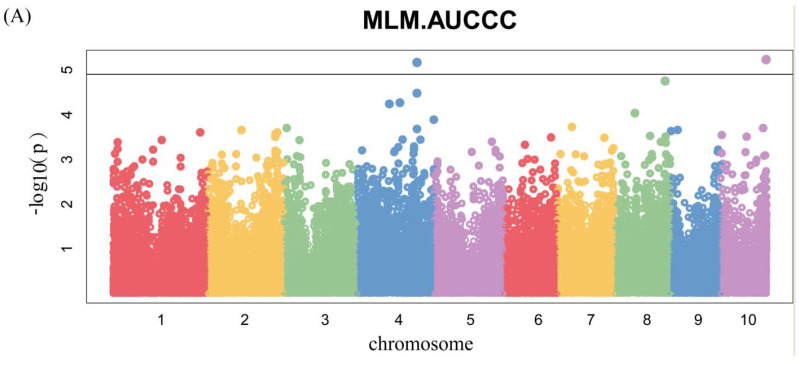
Genome-wide association studies of AUCCC. Each point represents an SNP site, and the horizontal black line represents that the effective threshold of Bonferroni correction at 1.03 × 10^−5^, different colors represent different chromosomes of maize. (**A**) FS; (**B**) LD; (**C**) BLUP.

**Figure 7 genes-14-01010-f007:**
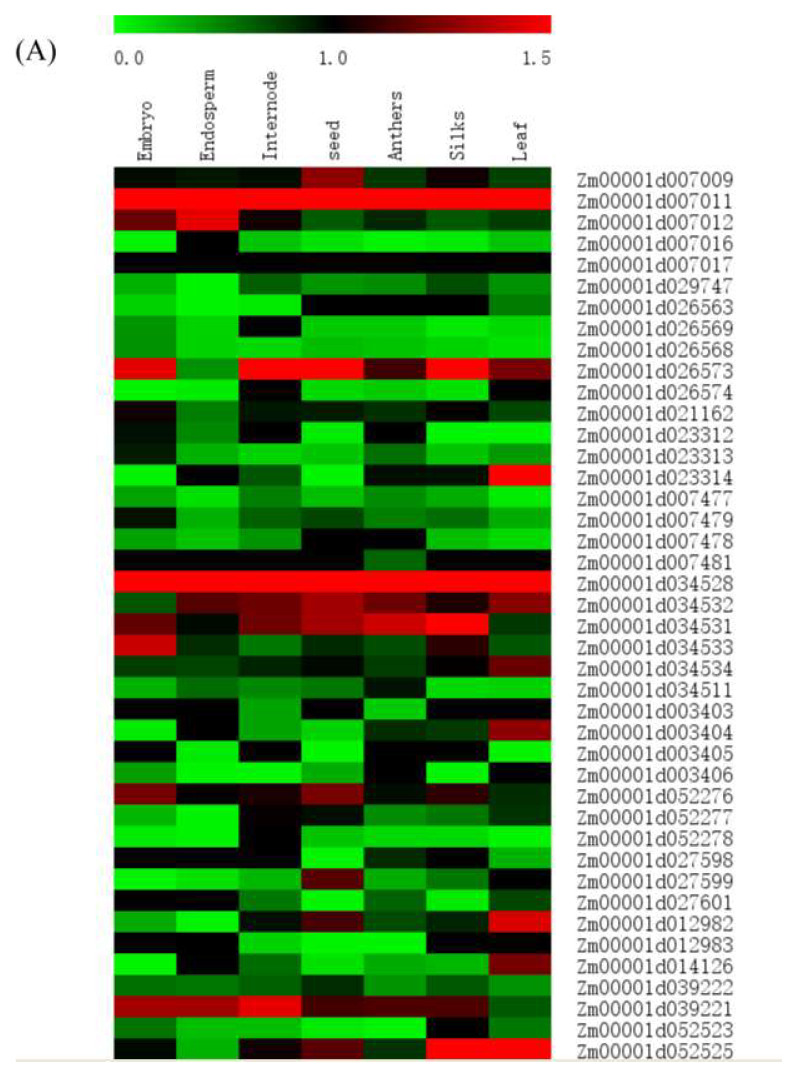
Heat map of the expression patterns of candidate genes determined by genome-wide association study. The value used in the figure is the log10 conversion ratio of the counts of standardized PRKM counts of chlorophyll content in (**A**) seven tissues and (**B**) in leaves on days 0, 6, 12, 18, 24 and 30 of maize silking stage. Columns and rows are sorted according to similarity. Compared with different periods of a specific gene, red, black and green represent higher, moderate and lower expression, respectively.

**Figure 8 genes-14-01010-f008:**
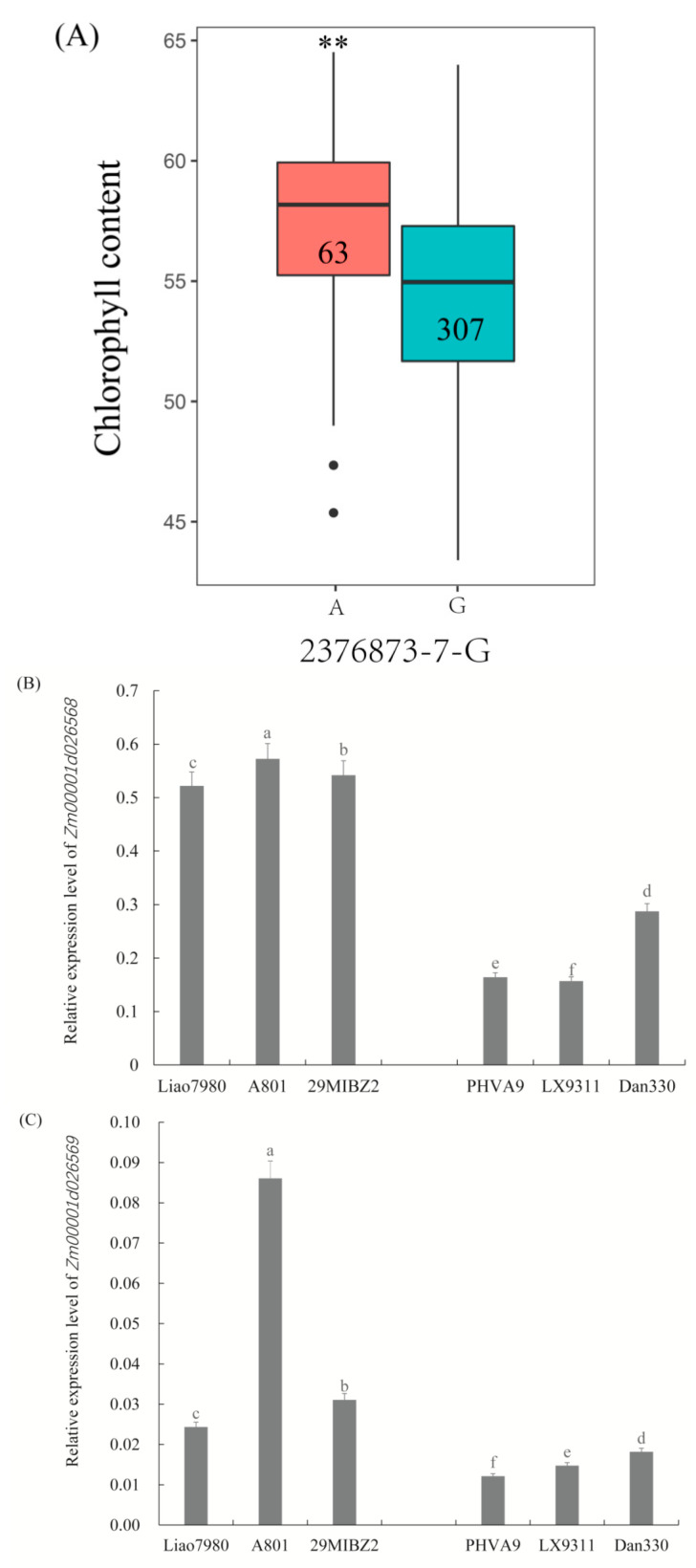
Allele variations of SNP and expression patterns of *Zm00001d026568* and *Zm00001d026569*. (**A**) Allele effects of SNP 2376873-7-G, 307 and 63, represent the number of G and A genotypes in the population, respectively. The relative expression of *Zm00001d026568* (**B**) and *Zm00001d026569* (**C**) by qRT-PCR in 6 lines, including Liao7980, A801 and 29MIBZ2, containing A allele; PHVA9, LX9311 and Dan330, containing G allele. Vertical bars indicate standard deviation. ** indicates statistical significance with *p* < 0.01. Bars with different letter superscripts are significantly different at *p* < 0.05.

**Table 1 genes-14-01010-t001:** Variance composition and broad-sense heritability of chlorophyll content and AUCCC in the maize association population in two environments (17FS and 17LD).

	Source of Variation ^a^	Mean Square	Significance ^b^	H^2 c^
chlorophyll content	Environment (E)	10058.20165	<0.01 **	0.67
	Genotype (G)	190.14580	<0.01 **	
	G × E	155.13709	0.4034	
AUCCC	Environment (E)	149.3227937	<0.01 **	0.66
	Genotype (G)	1.1585265	<0.01 **	
	G × E	0.405332	0.0987	

^a^ G × E indicates the interaction between G and E. ^b^ ** indicates statistical significance with *p* ≤ 0.01. ^c^ Family mean-based broad-sense heritability.

**Table 2 genes-14-01010-t002:** Single-nucleotide polymorphism (SNP) chromosomal positions and candidate genes significantly associated with chlorophyll content and AUCCC identified by genome-wide association study.

Trait	SNP	Chr	Position (bp)	*p* Value	Gene	Gene Interval (bp)	Annotation	Pathway
17BLUP chlorophyll content	Marker.247949	2	2.20 × 10^8^	9.10 × 10^−6^	*Zm00001d007009*	Chr2:220125041-220148355	DNAJ heat shock N-terminal domain-containing protein	Chloroplast targeting, photosystem II repair
					*Zm00001d007011*	Chr2:220152033-220155206	ATP synthase	Energy metabolism
					*Zm00001d007012*	Chr2:220170458-220175539	CHLOROPLAST RNA-BINDING PROTEIN	Chloroplast morphogenesis
					*Zm00001d007016*	Chr2:220204271-220219191	disease resistance protein RGA2	Disease-resistant
					*Zm00001d007017*	Chr2:220217348-220221222	thioredoxin-like protein AAED1 chloroplastic	Electronic circulation and daylighting
17FS chlorophyll content	2376873-7-G	10	1.48 × 10^8^	7.08 × 10^−6^	*Zm00001d026563*	Chr10:148153479-148157015	AP2/EREBP transcription factor 40	Plant growth and development
					*Zm00001d026569*	Chr10:148185247-148190302	chloroplastic palmitoyl-acyl carrier protein thioesterase	De novo synthesis of fatty acids
					*Zm00001d026568*	Chr10:148184610-148195615	pentatricopeptide repeat-containing protein	Chloroplast development
					*Zm00001d026573*	Chr10:148201459-148207853	5-methylthioribose kinase	Methylthioadenosine (MTA) cycle
					*Zm00001d026574*	Chr10:148208912-148217798	UDP-D-galacturonate	Homogalacturonan biosynthesis
17LD chlorophyll content	Marker.190636	2	43,281,907	6.97 × 10^−6^	*Zm00001d003403*	Chr2:43279990-43285413	aaap10—amino acid/auxin permease10	Amino acid transportation
					*Zm00001d003404*	Chr2:43284025.43288746	transmembrane protein	Transmembrane
					*Zm00001d003405*	Chr2:43285551-43289511	protease inhibitor/seed storage/LTP family protein	Signal transduction
					*Zm00001d003406*	Chr2:43298285-43304459	actin binding protein	Plant growth and development
	Marker.346487	4	8,724,268	8.32 × 10^−6^	*Zm00001d027598*	Chr1:8676135-8681000	cct101—CO CO-LIKE TIMING OF CAB1 protein domain101)	Transcription factors, floral completion
					*Zm00001d027599*	Chr1:8699800-8704356	alkane hydroxylase MAH1	Cuticular wax biosynthesis
					*Zm00001d027601*	Chr1:8772748-8777355	behenate ω-hydroxylase	Suberin monomers biosynthesis
	2504165-22-G	5	2,775,822	9.93 × 10^−6^	*Zm00001d012982*	Chr5:2770681-2776100	NAD(P)-binding domain containing protein	Energy metabolism
17LD AUCCC	Marker.571424	6	1.73 × 10^8^	7.40 × 10^−6^	*Zm00001d039222*	Chr6:173160699-173165510	cct40—CO CO-LIKE TIMING OF CAB1 protein domain40)	Transcription factors, floral completion
					*Zm00001d039221*	Chr6:173158717-173163744	nucleoside diphosphate kinase 4	Signal transduction
17FS AUCCC	2376873-7-G	10	1.48 × 10^8^	5.82 × 10^−6^	*Zm00001d026563*	Chr10:148153479-148157015	ereb40—AP2-EREBP-transcription factor 40	Plant growth and development
					*Zm00001d026568*	Chr10:148184610-148195615	pentatricopeptide repeat-containing protein, mitochondrial-like	Chloroplast development
					*Zm00001d026569*	Chr10:148185247-148190302	palmitoyl-acyl carrier protein thioesterase, chloroplastic	De novo synthesis of fatty acids
					*Zm00001d026573*	Chr10:148201459-148207853	5-methylthioribose kinase	Methylthioadenosine (MTA) cycle
					*Zm00001d026574*	Chr10:148208912-148217798	UDP-D-galacturonate	Homogalacturonan biosynthesis

## Data Availability

The original data for this article can be found online at: https://wp.me/p80aHo-L5.

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
