# Peer review of "Genome-Wide Association Study Identified Novel SNPs Associated with Chlorophyll Content in Maize"

_genes, 2023, doi:10.3390/genes14051010_

Round 1

Reviewer 1 Report

Comments to the author

The authors measured chlorophyll contents of ear leaves of maize plants at silking and several time points after pollination showing normal distribution of both chlorophyll contents and dynamic changes after pollination, suggesting that the phenotypes were controlled by genetic variations in the population. The authors performed GWAS analysis to identify SNPs correlating to chlorophyll content and dynamic changes in maize and identified a total of 19 SNPs and 76 candidate genes within the 50kb flanking region of these SNPs. Among them, the authors found that SNP 2376873-7-G with two candidate genes associated with chlorophyll contents and dynamic changes. In the end, the authors showed that the higher expression levels of these two candidate genes correlated with higher chlorophyll contents.

Overall, this manuscript is partially descriptive.

Major concerns:

1.      It is not clear how the authors prioritize the importance of the SNPs identified in this study or the effects of the SNPs on the chlorophyll contents in maize inbreds.

2.      Since chlorophyll content is a quantitative trait, there are several studies reporting QTLs for chlorophyll contents in maize and other crops. Do the authors find any SNPs or candidate genes from this study colocalize with QTLs that have been reported?

3.      The authors seem to focus only on SNP 2376873-7-G. Is there any specific reason? In addition, this SNP was identified in 17FS only but not in 17LD. Why?

4.      The authors verified the correlation of the expression levels of two candidate genes from the RNA extracted from seedlings at V4 stage. However, the SNPs were identified using the chlorophyll content in ear leaves of the plants at silking. Do the expression levels of these candidate genes maintain at the similar level in the leaves at V4 seedling stage and in the ear leaves at reproductive stage of the plant?

Below are more specific comments and suggestions.

Line 18: delete one “that”. There are two “that” in the sentence.

Line 24: rephrase this sentence to “higher expression levels of these two genes are associated with higher chlorophyll contents.”

Line 33: is an important source used for animal feed and bioenergy, and becomes the top one crop produced in China with the production estimated up to 272.6 million tons (MT) in 2021.

Line 44-47: need citation(s) to support this statement.

Line 52: the first step is the synthesis of chlorophyll….

Line 63: replace many with various.

Line 72: delete “tolerance” after soybean cyst nematode.

Line 73: tolerance of soybean cyst nematode and chlorophyll content were identified.

Line 90: Together, these studies indicate that GWAS is a reliable tool to study….

Line 114: delete “were made.”

Line 170: clearly indicate 17FS and 17LD is the section 2.2. Field experiments. Such as in May 2017 (17FS) and …….November 2017 (17LD).

Line 175: delete “which.”

Line 176: redo Figure 1 which is not properly labeled showing which figure means which set of data and what the numbers mean.

Line 180: * indicates statistical significance with P<0.05

Line 186: P<0.05 or 0.01? There are “**” shown in Figure 1.

Line 195: 10, 15, and 20 days after silking in 17FS, and at 0, 5, and 10 days after silking in 17LD.

Line 196: the highest at silking (day 0), and gradually decreased and showed the lowest at 20 days after silking during measurement…..

Line 211: please have the (A) and (B) on the top left corner of the figure panel for proper labeling and the same for all the figures.

Line 244: which SNP is this particular one? Is it important? Please specify why it is notable.

Line 261: replace “all in all” with “together”

Line 272: there is no horizontal black line showing the threshold in Figure 6C.

Line 278: in silico profiling

Line 289: it is not clear how the authors obtain the expression pattern of the candidate genes in leaves after pollination. It was not mentioned in the main text or in the material and method section. (From published RNAseq data or the authors performed RT-qPCR to measure the expression level?)

Line 290: after pollination.

Line 291: has low expression level at day 0.

Line 296: increased gradually after pollination during measurement.

Line 300: AAP2 gene was mutated in…

Line 301: than the wild-type plant.

Line 302: these results are in line with previous report.

Line 312: are associated with higher chlorophyll content

Line 314: how the relative expression levels of these two gene was obtained? By RT-qPCR? RNAseq?

Line 318: “C” genotype or “G” genotype?

Line 319: higher chlorophyll content was associated with higher expression levels of these two candidate genes in the selected lines shown in this study.

Line  332: is there any significant difference in the chlorophyll content shown in Figure 8A?

Reviewer 2 Report

Dear Author,

The current manuscript deals with a genome-wide association analysis of chlorophyll content in maize, which shows interesting results. The manuscript is well-written and well-presented. The main concern is that there was a similar study on the same topic published in 2017 (DOI :10.13560/j.cnki.biotech.bull.1985.2017.04.013) and entitled "Genome-wide Association Study of Chlorophyll Content in Maize Leaves" in the journal Biotechnology Bulliten. The author is requested to indicate if the current study was performed on the same maize inbred lines and highlight the importance of the findings of the current study in relation to their findings. Also, in the discussion part, the author elaborated on a description of results; therefore, a little more discussion is needed specifically on the validation of candidate gene expression using an in silico approach.

There are a few minor things in the text:

* Using the expression "was had" is confusing.

* The abbreviation "17FS" means what?

* What does the author mean by " This result may be caused by different species"? (line 223)

* Line 237:  author wrote, "two were on chromosome 2, 4, 10, respectively." please correct!

Round 2

Reviewer 1 Report

Thank you for addressing most of my concerns. However, there is still several points that the authors did not explain clearly.

In the major concern 4:

The authors examined the relative expression levels of two candidate genes from 6 different inbred lines. In the method section 2.7, the authors clearly pointed out that the RNA was extracted from the leaves of maize plants at V4 stage and transcription levels were analyzed using RT-qPCR. Yet, the authors responded to the question that the RNA was not extracted from seedling at V4 stage which does not make any sense.

1) Is the expression level of these two candidate genes in the leaves at V4 stage comparable to the ear leaves at silking?

2) Is the expression level of the genes highly correlated to the chlorophyll contents in both leaves at V4 stages and ear leaves during silking?

From Line 316 to 318: 

In contrast, the relative expression of pentatricopeptide repeat-containing protein and chloroplastic palmitoyl-acyl carrier protein thioesterase was lower in lines carrying the "C" genotype (Figure 8B-C).

There is no "C" genotype!
